## COMMENT

# Translating area-based conservation pledges into efficient biodiversity protection outcomes

Charles A. Cunningham [1]✉, Humphrey Q. P. Crick [2],
Michael D. Morecroft [3], Chris D. Thomas [1,5]✉ & Colin M. Beale [1,4,5]✉

Ambitious national and global pledges to protect increasing areas of land risk trading conservation effectiveness for convenience of designation. We show that UK conservation areas often lie outside the highest biodiversity priority landscapes, and that systematic conservation planning can improve site selection.

National commitments under the Convention on Biological Diversity (CBD) have repeatedly under-delivered: global biodiversity indicators continue to decline[1] and the Aichi target 11 to protect 17% of the global terrestrial area by 2020 has not quite been met, with coverage currently standing at around 16.64%[2,3]. As elsewhere, the UK's 2010 commitments to halt biodiversity loss by 2020 have not been realised[4]. Globally, the response of conservationists and policymakers to these failed targets has been to propose ever more ambitious targets as we move towards the post-2020 global biodiversity framework[5,6]. Thus, the CBD has drafted a proposal to ensure that, by 2030, at least 30% of global land and sea are conserved, "especially areas of particular importance for biodiversity and its contributions to people"[5]. However, there is a risk that states will then designate land to maximise 'apparent protection', and not necessarily outcomes for biodiversity[7].

### The UK 30by30 pledge

In this context, the British Prime Minister announced a new commitment on the 28th September 2020 to protect 30% of the UK's land by 2030 to support the recovery of nature[8]. This extends to the terrestrial environment the existing '30by30' pledge to protect 30% of British seas by 2030[9]. The potential for such pledges to prevent biodiversity loss will depend on the extent to which targets are met, and whether they are met in a way that delivers effective conservation outcomes[10].

Newly designated protected areas or other effective area-based conservation measures (OECMs) should complement the existing network of conservation sites if they are to maximise the representation (and thereafter protection) of species[11]. Currently, only 9.04% of Britain's land area has a legal status that specifically mandates biodiversity protection, equivalent to IUCN level IV[12]. The British Prime Minister's 30by30 pledge also includes an additional 17.67% of land that is currently designated as 'protected landscapes', such as National Parks and Areas of Outstanding Natural Beauty, which are classed as lower-grade IUCN level V protection[12]. They are multi-purpose landscapes with a focus on planning and development constraints that do not confer additional legal protection for wildlife (above any national legislation that applies to all land, or additional biodiversity designations at specific locations within these protected landscapes). Thus, two-thirds of the land that has been identified as contributing to the 30% pledge has neither been selected to protect important biodiversity, nor offers specific protection to

[1]Leverhulme Centre for Anthropocene Biodiversity, University of York, York, UK. [2]Natural England, Eastbrook, Cambridge, UK. [3]Chief Scientist's Directorate, Natural England c/o Natural England Mail Hub, Worcester, UK. [4]York Environmental Sustainability Institute, University of York, York, UK. [5]These authors contributed equally: Chris D. Thomas, Colin M. Beale. ✉email: cac567@york.ac.uk; chris.thomas@york.ac.uk; colin.beale@york.ac.uk

biodiversity. In order to reach the 30% goal, a further 3.29% of the land surface outside these sites still requires protection.

As is characteristic of ambitious conservation aspirations, delivering nature recovery in practice is far from straight-forward. In densely populated countries like the UK and elsewhere in Europe, priority species are often confined to small habitat fragments[13]. This makes it hard to establish the landscape-scale protection and restoration of nature that is necessary if long-term species survival is to be ensured[14]. Area-based conservation priorities should thus focus on locations where a combination of extending and managing existing sites, improving marginal habitats nearby, restoring additional habitats and improving landscape-scale connectivity are most likely to be effective[15]. The likelihood that individual threatened species will recover would be increased in these areas because they are already present, and thus available to colonise improved habitats that are delivered by upgraded protection and management in the surrounding landscape[14,15]. To inform this expansion, we explore alternative scenarios to identify the highest conservation priority locations in Great Britain. We identify priority areas that currently fall outside of national biodiversity designations (minimum IUCN level IV protection) and, separately, those that fall outside biodiversity designations and protected landscapes combined (minimum IUCN level V protection). We deduce how well these strategies deliver species conservation priorities in 30% of Britain's land area (see Supplementary Fig. 1 for analysis workflow).

## Achieving 30% land coverage with systematic planning

The best outcomes for biodiversity are expected when priority sites are selected (and conservation measures implemented) on the basis of the species or habitats on all sites, unconstrained by historic conservation decisions. In practice, sites currently protected primarily for biodiversity are very unlikely to lose their protection in the UK, so our first scenario (scenario 1, Fig. 1a) represents a systematic conservation prioritisation that includes all the sites currently protected for biodiversity. We identified the highest priority areas for network expansion that maximises coverage of 445 priority species distributions including birds, plants and a wide variety of invertebrates (Supplementary Data 1). An important additional consideration is the existing land use of the cell[16,17], and so we also undertook a parallel analysis incorporating opportunity costs of protecting or restoring land using an agricultural/urban land classification (Supplementary Fig. 2; Supplementary Table 4). Note that prioritisations are undertaken at the 10 × 10 km scale (henceforth 'cells') due to the resolution of spatial data for certain taxa. Attaining 30% national coverage by protecting *all* the land within selected 10 × 10 km cells is not practical as most British landscapes have fragmented semi-natural habitat. The priority cells recognised here represent foci for identifying and directing subsequent conservation actions and funding, accepting that different blends of conservation actions will be required in different landscapes. Given this constraint, an additional 50% coverage target is presented, within which a subset of higher-priority sites can be

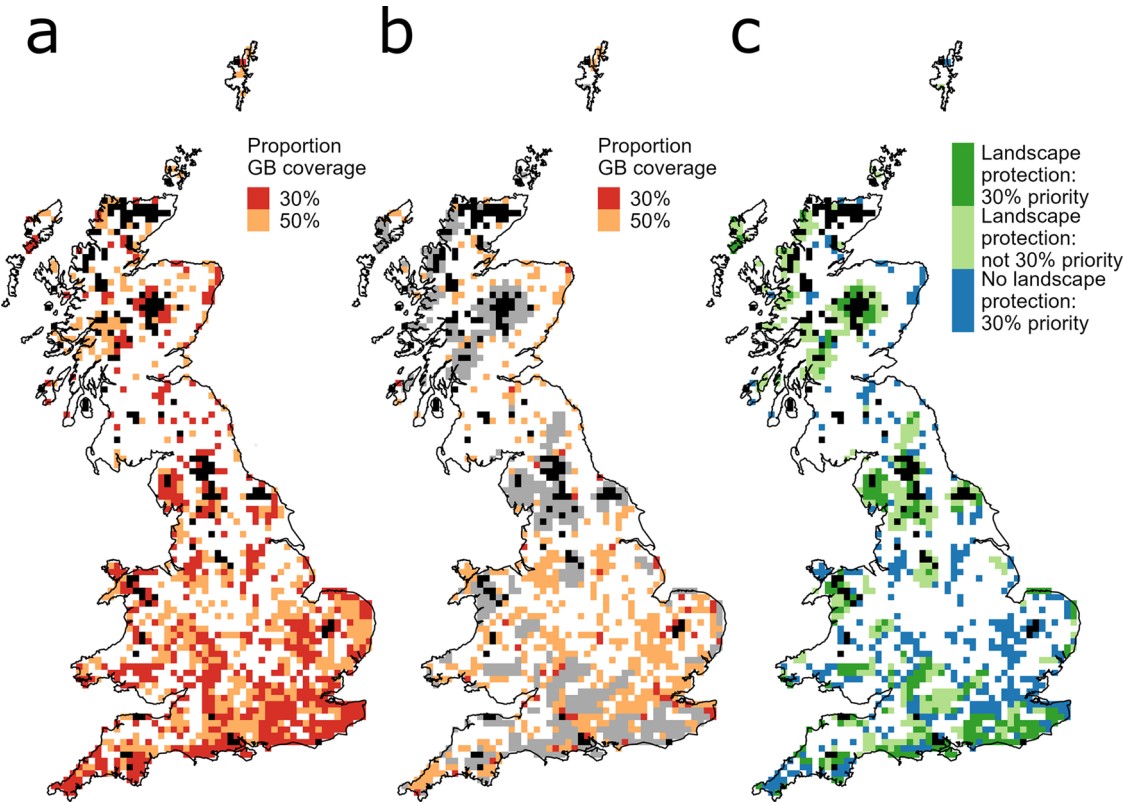

**Fig. 1 Spatial priorities for additional area-based conservation to meet 30by30 coverage targets within Britain. a** Scenario 1: prioritisation constrained only by the inclusion of current biodiversity protected sites. **b** Scenario 2: constrained by maintaining both biodiversity and landscape protection sites, as suggested by the 30by30 announcement. **c** Overlap between top 30% priority cells for biodiversity from scenario 1 and current protected landscapes. Cells already protected for biodiversity are shaded black (which are included as part of the 'top 30% in both scenarios). For panels (**a**) and (**b**), top 30% priority cells are shaded red, top 50% orange, and landscape protection cells are grey. In panel (**c**), priority cells for biodiversity are dark green if in a landscape protection cell and dark blue if outside a landscape protection cell; light green shows those landscape protected cells that are not a priority for biodiversity conservation.

identified as foci for biodiversity and habitat 'recovery' in the wider countryside (Fig. 1, Supplementary Table 2).

Under scenario 1, the most important areas to prioritise for attaining at least 30% network coverage, in a way that is likely to benefit the most species, are largely concentrated in southern and eastern England (Fig. 1a), although priority cells were less concentrated in the south if land (opportunity) costs were included (Supplementary Fig. 3a). Northern and upland areas of Britain have disproportionately larger areas protected for biodiversity[14], so the greatest gains in species representation can potentially be achieved by increased levels of protection and habitat restoration in southern and lowland areas.

### Achieving 30% land coverage with pledged landscapes

In a second analysis, we identified spatial conservation priorities when constrained by including both biodiversity and landscape protection cells (scenario 2, Fig. 1b, Supplementary Fig 3b). In line with the 30by30 pledge, this scenario additionally includes all current protected landscapes, and we identify further priorities to expand the network to achieve 30% coverage. Under this scenario, cells with the highest priority are again scattered primarily in southern England (Fig. 1b), but are again more spread when opportunity costs are considered (Supplementary Fig. 3b). Both scenarios would protect more of the ranges of threatened species than the cells currently protected for biodiversity (median 1.63% distribution protected): the less constrained first scenario would ensure an additional 59.54% could be protected within 30% of cells in scenario 1, compared to 37.69% under scenario 2 (Fig. 2, Supplementary Table 2). The latter comprises 29.47% from existing biodiversity and landscape protection (in 27.80% of national cells), with the additional prioritised land contributing the extra 9.85% (in 2.20% of national cells). This is only slightly more effective than undertaking scenario 2 by replacing landscape protection cells with the same number of randomly sampled cells (mean 32.48%, min 30.04%, max 34.30%; based on 10,000 iterations).

The higher representativeness of scenario 1 reflects the fact that 62.39% of priority 30% cells in scenario 1 fall outside currently protected landscape cells (Fig. 1c blue, Supplementary Table 3), and just 4.77% of the land within these cells is already protected

for biodiversity (mostly as small individual reserves). These are regions where new area-based biodiversity conservation would bring greatest rewards *outside* protected landscapes. In contrast, only 41.50% of protected landscape cells lie within scenario 1 priority 30% cells (Fig. 1c dark green, Supplementary Table 3). These form the highest priority cells for upgrading biodiversity conservation *within* protected landscape cells: current protection for biodiversity (at higher level IUCN level IV designation) is only 10.27% of the total area within these protected landscape cells. This indicates that to meet the 30by30 target efficiently, biodiversity protection would need to be targeted in a subset of the protected landscapes as well as in additional areas outside protected landscapes. The planned Nature Recovery Network provides an opportunity to implement this, potentially including 25 catchment or landscape-scale Nature Recovery Areas in currently non-designated areas, as well as creating/restoring 500,000 ha of new priority habitat[18].

### Making conservation pledges deliver for nature

The 30by30 commitment is a positive step for UK conservation, but requires detailed planning and implementation if it is to deliver its intended goals. Careful targeting of new area-based conservation is required to maximise biodiversity representation, with protection and management needed to ensure that priority species (and other beneficial features of the landscapes) are not lost, and that populations can subsequently expand into the surrounding landscapes. These conservation goals will be met more efficiently if prioritisation occurs with the fewest possible constraints. However, if protected landscapes (National Parks, Areas of Outstanding Natural Beauty, Scottish National Scenic Areas) are included in the 30% coverage target, the impact on rare species will be limited unless habitats are improved within them, as well as carefully targeting the extension of the conservation network beyond currently designated landscapes (Fig. 1). Further development of priority conservation networks should consider how climate change will likely affect the distribution of species, habitats, and land use pressure[19,20], but securing the existing distributions of currently threatened species remains a priority. As more ambitious area-based conservation targets are likely to be adopted by other states as part of the post-2020 global

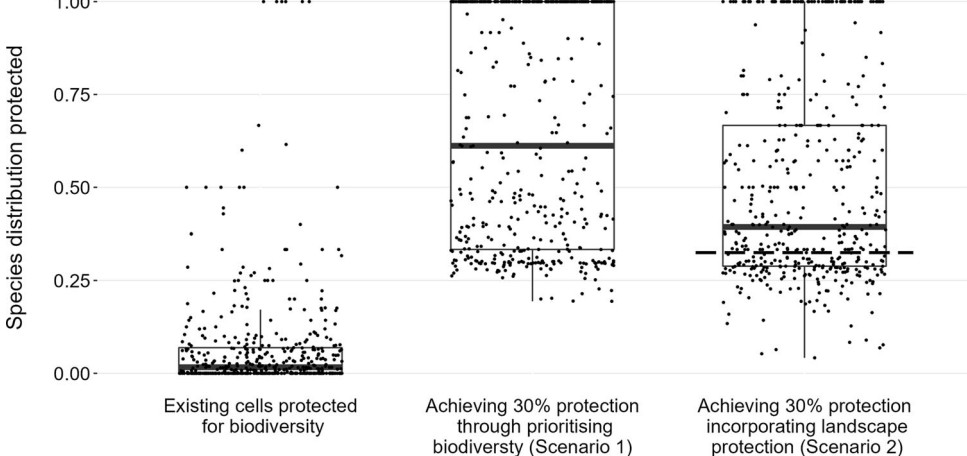

**Fig. 2 Proportion of 445 species distributions covered by different conservation scenarios.** Cells were considered currently protected for biodiversity if >40% of the cell was designated IUCN level IV land or higher (6.41% of national cells). Scenario 1 involved attaining 30% GB cell coverage by maximising proportion of species distributions covered, constrained only by the inclusion of current cells protected for biodiversity. Scenario 2 was constrained by maintaining cells protected for biodiversity along with additional protected landscape cells (27.80% of national cells), as suggested by the 30by30 announcement. The lower and upper borders of the box are first and third quartiles, respectively; the horizontal bar is the median; and whiskers extend to 1.5 * inter-quartile range. Individual species are overlaid as points. The dashed line on scenario 2 shows the average of 10,000 sample medians where a randomly selected equivalent number of cells were incorporated instead of landscape protection, before prioritisation.

biodiversity framework, our analysis exemplifies how important it is that such areas are chosen for their ability to deliver efficiently and effectively for biodiversity, given that there are increasing demands on land for a wide range of other uses.

## Methods

All analysis was undertaken in Great Britain and associated islands over 20 km². All prioritisations were undertaken at a $10 \times 10$ km landscape-scale on cells with greater than half land coverage. We considered designations 'protected for biodiversity' to be Sites of Special Scientific Interest (SSSI) and National Nature Reserves (NNR); and landscape protection designations to include National Parks (NP), Areas of Outstanding Natural Beauty (AONB), and Scottish National Scenic Areas (NSA). Different cell protection 'cutoffs' were tested at 30, 40, 50, 60, and 70% (Supplementary Table 1). Hence cells were considered to be 'protected for biodiversity' at the landscape-scale if SSSI/NNR coverage was above the percentage land cutoff, e.g. at least 40% IUCN IV protection (Fig. 1: black cells). 'Protected landscapes' were $10 \times 10$ km cells with total coverage from all of the designations above the cutoff, e.g. at least 40% IUCN V (or greater) protection, but under 40% level IV protection (Fig. 1b: grey cells). Results were qualitatively similar for all cutoffs (Supplementary Tables 2 and 3). The joint proportion of cells protected for biodiversity and protected landscapes were most similar to the actual coverage at the 40% 'cutoff' (27.80% of $10 \times 10$ km cells 'protected' compared to 26.71% actual area coverage), and this is presented in the main text. All designation data used is publicly available from the respective national spatial data repositories for England[21] (SSSI/NNR/NP/AONB), Scotland[22] (SSSI/NNR/NP/NSA), and Wales[23] (SSSI/NNR/NP/AONB).

We used the recorded distributions of 445 priority species listed under the Section 41 (Natural Environment and Rural Communities Act, 2006), provided by Butterfly Conservation (BC), Biological Records Centre (BRC); and breeding bird atlas data from British Trust for Ornithology (BTO)[24]. BTO bird atlas data are only available at the $10 \times 10$ km scale, which limited the spatial resolution of the analysis. We used all priority species that we were able to acquire from the above recording bodies between 2000 and 2014 (Supplementary Data 1). We used the raw distribution records for 156 species that were very localised (10 or fewer presence records) and for a further 77 species which could not be modelled (most of which were also very rare, and for which models did not converge). For the remaining 212 species with over 10 presence records, we interpolated their range using Integrated Nested Laplace Approximations (INLA) in the *inlabru* R package[25]. We used a joint model predicting distribution while accounting for recording effort, including biologically relevant covariates: seasonality, growing degree days, water availability, winter cold[26], and soil pH from the Countryside Survey 2007 dataset[27]. These covariates were calculated from monthly means of weather data (mean temperature, sunshine and rainfall) for the decade to 2014 provided by the Met Office[28]. We also included soil moisture in the calculation of water availability[29]. We used raw data records from all 445 species, along with broad habitat layers extracted from the Land Cover Map 2015[30], in a Frescalo analysis[31] to estimate recorder effort. See Supplementary Methods for further details of modelling.

We carried out a spatial prioritisation using Core Area Zonation[32], whereby cells are removed iteratively, first removing those that contribute the smallest cell value: the maximum proportion of species distributions within the remaining cells. In this way cells remaining longer within the solution complement species representation of other cells to a greater extent, and hence contribute most to underrepresented species' distributions.

However, priorities were constrained by masking or 'locking in' different relevant areas to each scenario such that all other cells must be removed first; reducing overall solution optimality but ensuring complementarity to masked areas. Scenario 1 only masked cells protected for biodiversity and didn't consider other designations beyond that. Scenario 2 also masked cells protected for biodiversity but, corresponding to the 30by30 pledge, additionally masked protected landscapes.

We undertook a parallel analysis additionally incorporating opportunity costs calculated from agricultural land classification and urban areas[33–35] (Supplementary Fig. 2, Supplementary Table 4). Although urban areas are often excluded from SCP analyses, it is important to consider species complementarity of all landscapes (the government 30% target applies to the entire land surface). Since some urban/near-urban areas contain nationally rare species, we include urban areas, albeit imposing the maximum opportunity cost in these cells. In this analysis, cell value was calculated as the maximum proportion of species distributions within the remaining cells divided by the mean opportunity cost of the cell (Supplementary Fig. 3, Supplementary Tables 2 and 3).

**Reporting summary**. Further information on research design is available in the Nature Research Reporting Summary linked to this article.

## Data availability

The spatial designated sites data that were used to create the masks are available from public repositories (England https://naturalengland-defra.opendata.arcgis.com, Scotland https://spatialdata.gov.scot/geonetwork/srv/eng/catalog.search#/home and Wales https://lle.gov.wales/catalogue?lang=en). The agricultural land classification data used to calculate opportunity costs are similarly publicly available (England https://data.gov.uk/dataset/952421ec-da63-4569-817d-4d6399df40a1/provisional-agricultural-land-classification-alc, Scotland https://www.hutton.ac.uk/learning/natural-resource-datasets/landcover/land-capability-agriculture and Wales http://lle.gov.wales/catalogue/item/PredictiveAgriculturalLandClassificationALCMap2/?lang=en). The species record data that were interpolated and included within the scenario prioritisations are available from BRC, BC and BTO but restrictions apply to the availability of these data from BRC and BC, which were used under license for the current study, and so are not publicly available. Modelled data are however available from the authors upon reasonable request and with permission of BRC and BC. Prioritisation masks and ranks used to create the figures of this text are accessible through Figshare[36].

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

## Acknowledgements

This work was funded by the Natural Environment Research Council (grant NE/R012164/1), with additional support from Natural England. C.D.T. thanks the Leverhulme Trust for financial support through a Research Centre grant (RC-2018-021). We thank the many recording schemes which provided data used in this analysis; the Biological Records Centre (supported by Natural Environment Research Council - grant NE/R016429/1), British Trust for Ornithology (BTO), Butterfly Conservation, and Hoverfly Recording Scheme (HRS), and to the committee and members of the Bees Wasps and Ants Recording Society (BWARS) for permission to use their data – downloaded on 24.05.2018. We also thank the two anonymous reviewers for their constructive comments on the manuscript.

## Author contributions

C.D.T. and C.M.B. initially conceptualised the project. C.M.B. and H.Q.P.C. secured financial support for the work. C.A.C. carried out the spatial analysis and prioritisations with input throughout from C.D.T., C.M.B., H.Q.P.C., and M.D.M. All authors contributed to the final manuscript.

## Competing interests

The authors declare no competing interests.
