## [Peer Review File · Communications Biology]

Reviewers' comments:

Reviewer #1 (Remarks to the Author):

This paper addresses the recent commitment by the UK government to protect 30% of land by 2030, in line with the global '30by30' commitment of the 2030 Agenda for Sustainable Development. The authors expand the current protected area network in the UK, under 2 baseline 'protection' definitions, to reach this 30% target to maximize representation of priority species distributions. They found that expanding the protected area network using the strictest protected areas as a baseline would yield the maximum representation of threatened species—a strategy that includes adding new protected areas and upgrading current protected areas with lower-level mandates for biodiversity protection.

While systematic conservation planning will be critical to achieving the 30by30 target, the authors fail to account for numerous factors needed to achieve "careful targeting of new area-based conservation." (99) The authors claim their analysis "exemplifies how important it is that such areas are chosen for their ability to deliver efficiently and effectively for biodiversity, given that there are increasing demands on land for a wide range of other uses." (108-111) But their analysis does not provide any evidence that the proposed protected area prioritizations would be efficient or effective, as there is no assessment of connectivity, fragmentation, local or surrounding land uses, or costs associated with protection in different areas. That said, it would be a shame for the authors to miss such an opportunity for creating real policy impact—which could be done with significant modifications. I therefore suggest major revisions be undertaken in order to improve the manuscript.

MAJOR COMMENTS

1. The authors are not contributing what they think they are contributing. The networks they have prioritized are maps of areas most representative of current species distributions. While that does, of course, have value, it is not a map of the most efficient and effective areas for protection under the 30by30 target.

To save myself some time, I refer the authors to Brown et al. (2015). The authors ignore the key principle of systematic conservation planning: maximizing return on conservation investment. The current analysis is not guided by any objective—no minimum representation target, no consideration of protection costs, no consideration of existing land use/land configuration. The authors state they "deduce how well these strategies deliver efficient, representative 30by30 priorities" – but they do not assess the efficiency of their network, just species representation.

The authors should either (1) revise their analysis to prioritize a protected area network that (e.g.) meets a minimum representation target, while minimizing acquisition costs of land that is undeveloped/suitable for protection; or (2) reframe the manuscript to better reflect the analysis, which is a map of the 30% of the UK with greatest species representation potential (without claiming that this network is a proper SCP for the 30x30 target).

C. J. Brown et al. (2015) Effective conservation requires clear objectives and prioritizing actions, not places or species. PNAS 112: E4342.

2. There are several oddities in the methodological approach that either need remedied or need appropriate justification.

The authors use a 40% threshold for locking in protected areas (Line 120). This is a very strange threshold that I have never seen in SCP applications. Typically, a 50% threshold is used because it is, in fact, the majority of the cell. Why did the authors consider an entire cell to be 'protected' if only 41% of it was protected? Without references to other papers establishing this as a common practice (which maybe I'm unaware of) or a context-specific justification, this feels arbitrary at best and inaccurate at worst.

I don't understand why the authors are using 100 km² planning units for such a (relatively) small

case study, where the dense population has led to small fragmented habitats that could be available for protection. The authors recognize that it's unrealistic to expect 100% of land within these cells would be protected, but their solution is to expand the area target to 50% of cells. This is problematic for multiple reasons, as this avoids any actual reporting of the land available to be protected, and they don't even report on the results of the 50% (except for its inclusion in the figure).

As a result, it is ambiguous just how much of the 30% of cells chosen for protection actually contain land that would be protected. Furthermore, they do not appear to exclude developed lands from potential selection. A cell chosen for protection could be 80% built-up areas and yet 100% of it contributes to their prioritized areas for protection.

The authors should remove developed land from consideration (unless with proper justification), and they should reduce the size of their planning units to more adequately reflect the highly fragmented state of UK habitats so they can more accurately estimate area coverage and assess true connectivity of their networks.

It is also strange that the authors do not consider range shifts amidst climate change. It is becoming very apparent that new protected areas need to protect not just what biodiversity features are present, but what biodiversity will be present in the future. This is becoming common practice in systematic conservation planning, yet the authors claim that protecting what's "present" (Line 47) enhances the chances that species will recover and colonize the landscape. But that is not the optimal approach to securing species from extinction. The authors should consider adding expected shifts due to climate change, or provide good justification for why they are not considering it in their proposed protected areas.

3. There is a lack of reproducibility as the manuscript current stands. Since I have not received (nor seen any reference to) any supplementary material, I assume none exists. Without more detailed methods, I could not reproduce this analysis.

The authors do not state what species were included in the analysis. Did this include mammals, birds, insects, plants? There are also no references in the reference list to the species data. They mention the number of species with limited records, but they should provide a list of all species included in the analysis, and which ones were treated differently in the analysis.

The authors are too vague in their description of the analysis. For example they mention using a Frescalo analysis, where they apply weights for spatial proximity and habitat similarity (Line 138-140). How are the variables treated in the analysis, and what weights were applied? Simply stating that weights were applied does not tell me anything about how to reproduce it (or even verify if weights were applied properly). Another example in Line 136, the authors say means of weather data were used to calculate "relevant covariates", but I have no idea what "relevant covariates" mean.

The authors mention in Line 82 a random sample was taken, but in Line 175 they say 1000 random samples were drawn. Is it 1 or 1000 random samples? If it is the latter, the authors should also present the distribution of these results to compare with their scenarios. This is never mentioned in the Methods.

Reviewer #3 (Remarks to the Author):

Dear Editor,

The manuscript COMMSBIO-20-3548-T "Translating area-based conservation pledges into efficient biodiversity protection outcomes" is a short communication. It focuses on the issues that area-based conservation pledges without planning can bring to biodiversity conservation. It uses the case of United Kingdom to illustrate such issues. In UK there is a political commitment of protect 30% of the land by 2030 to support the recovery of nature. The UK's government assumed to

reach that 30% by dedicating 4% of new lands to the existing protected land network (26%). The authors claim however that such area-based pledges should be driven by the distribution of biodiversity (list of priority species). They shown how systematic conservation prioritization could improve the amount of biodiversity protected (species) and how it would result in a different conservation network regarding to nowadays. They combined advanced modelling tools to test a species-driven area-based conservation pledge under different land governance scenarios.

2. Overall impression of the work

The work is very pertinent for scientific community and in general to countries and society. Many countries have done area-based conservation pledges, but their efficiency in provide biodiversity protection outcomes remain poorly documented. The authors make an effort to advance discussion in this topic.

It is pleasant to read the manuscript. There is however space for improvements, namely by:

Main text

- The content is well organized.
- To access the numbers presented, the maps included may not be sufficient. The manuscript would be improved by including graph/summary statistics representations for the numbers presented (e.g. Line 84-92). These percentages are in base to what extent (e.g. km²).
- What type of species are included in the work? Animals, plants? There is no indication about that. This information can help us to understand if the work covers the broad biodiversity or is for instance driven only by e.g. animal species. This information is relevant for the readers and for the overall statement "Translating area-based conservation pledges into efficient biodiversity protection outcomes"

Methods

- Including a workflow showing the steps performed, assumptions made, input data used and connections between outputs. For a more common reader, this would benefit the understanding.
- Adding additional details in the modelling performed could be also submitted as attachment to support the robustness and clearance of the modelling process.
- mentioning all datasets used (e.g. Where comes from the data used to select all cells with fractional land cover > 60%, considered later as suitable cells for conservation expansion.)
- describing better in the methods the scenarios briefly introduced in the main text. They are important assumptions for the outcomes.

3. Specific comments, with recommendations for addressing each comment

- 1- Find and replace double spaces along the text.
- 2- Please make some sentences shorter and more clear for general readers (e.g. line 76-80 "In terms of representativeness, both scenarios would protect more of the ranges of threatened species than the cells currently protected for biodiversity (median 1.67% distribution protected): the less constrained first scenario would ensure an additional 68.27% could be protected within 30% of cells in scenario 1, compared to 38.33% under scenario 2 (Fig 2)")

Translating area-based conservation pledges into efficient biodiversity protection outcomes

(Commun. Biol)

Revision overview

We have undertaken a substantial additional analysis incorporating costs based on agricultural land value. We repeat the entire analysis, this time including costs, and the main results are very similar. The UK policy is currently stated in terms of an area commitment and nature 'recovery', and hence costs are not explicitly included within existing policy statements. Thus, we retain the original analyses in the main text, but agree that it is valuable to include equivalent cost-linked analyses as supplementary results. A number of supplementary figures and tables have also been added to fully address the constructive reviewer comments.

N.B.

- Colour on Figure 1c has been changed to make interpretation easier
- Number of randomised samples taken in evaluating scenario 2 (equivalent to protected landscape cells) is now 10 000

Comments are repeated in italics, and addressed in bold text either by a description of how they are adopted, or if not then an explanation. To illustrate how comments have been addressed, direct quotes are sometimes used from the manuscript.

Reviewers' comments:

Reviewer #1 (Remarks to the Author):

This paper addresses the recent commitment by the UK government to protect 30% of land by 2030, in line with the global '30by30' commitment of the 2030 Agenda for Sustainable Development. The authors expand the current protected area network in the UK, under 2 baseline 'protection' definitions, to reach this 30% target to maximize representation of priority species distributions. They found that expanding the protected area network using the strictest protected areas as a baseline would yield the maximum representation of threatened species—a strategy that includes adding new protected areas and upgrading current protected areas with lower-level mandates for biodiversity protection.

While systematic conservation planning will be critical to achieving the 30by30 target, the authors fail to account for numerous factors needed to achieve "careful targeting of new area-based conservation." (99) The authors claim their analysis "exemplifies how important it is that such areas are chosen for their ability to deliver efficiently and effectively for biodiversity, given that there are increasing demands on land for a wide range of other uses." (108-111) But their analysis does not provide any evidence that the proposed protected area prioritizations would be efficient or effective, as there is no assessment of connectivity, fragmentation, local or surrounding land uses, or costs associated with protection in different areas. That said, it would be a shame for the authors to miss such an opportunity for creating real policy impact—which could be done with significant modifications. I therefore suggest major revisions be undertaken in order to improve the manuscript.

We thank the reviewer for their substantial comments and have undertaken a number of additional analysis, supplementary information, and reworking of text to address these.

MAJOR COMMENTS

1. The authors are not contributing what they think they are contributing. The networks they have prioritized are maps of areas most representative of current species distributions. While that does, of course, have value, it is not a map of the most efficient and effective areas for protection under the 30by30 target.

To save myself some time, I refer the authors to Brown et al. (2015). The authors ignore the key principle of systematic conservation planning: maximizing return on conservation investment. The current analysis is not guided by any objective—no minimum representation target, no consideration of protection costs, no consideration of existing land use/land configuration. The authors state they "deduce how well these strategies deliver efficient, representative 30by30 priorities" – but they do not assess the efficiency of their network, just species representation.

The authors should either (1) revise their analysis to prioritize a protected area network that (e.g.) meets a minimum representation target, while minimizing acquisition costs of land that is undeveloped/suitable for protection; or (2) reframe the manuscript to better reflect the analysis, which is a map of the 30% of the UK with greatest species representation potential (without claiming that this network is a proper SCP for the 30x30 target).

C. J. Brown et al. (2015) Effective conservation requires clear objectives and prioritizing actions, not places or species. PNAS 112: E4342.

We have undertaken the first method here, revising the analysis by incorporating opportunity costs into the prioritisation analysis (L64-67; L173-176; Fig S1, S2; Table S5):

“An important additional consideration is the existing land use of the cell^{15,16}, and so we also undertook a parallel analysis incorporating opportunity costs of protecting or restoring land using an agricultural/urban land classification (Fig. S2; Table S5).”

The objective is clear, to maximise species representation while minimising costs. We do not set a minimum representation target as we argue this would not make sense in the context of the prioritisation: the coverage target has already been set by 30% GB coverage, we are seeking to allocate this in the most efficient way.

2. There are several oddities in the methodological approach that either need remedied or need appropriate justification.

The authors use a 40% threshold for locking in protected areas (Line 120). This is a very strange threshold that I have never seen in SCP applications. Typically, a 50% threshold is used because it is, in fact, the majority of the cell. Why did the authors consider an entire cell to be ‘protected’ if only 41% of it was protected? Without references to other papers establishing this as a common practice (which maybe I’m unaware of) or a context-specific justification, this feels arbitrary at best and inaccurate at worst.

We agree this was not clearly addressed in the text. 40% cell protection threshold was chosen as it aligns the proportion of GB hectads protected (27.79%), with the actual proportion of GB protected (26.71%) (Fig. S1). To address this arbitrary selection, we also carry out the analysis for 30%, 50%, 60%, and 70% cutoffs (Table S1, S2, S3) and find similar results for all. Added text L135-143:

“Different cell protection ‘cutoffs’ were tested at 30%, 40%, 50%, 60%, and 70% (Table S1).”

“The joint proportion of cells protected for biodiversity and protected landscapes were most similar to the actual coverage at the 40% ‘cutoff’ (27.78% of 10x10 km cells ‘protected’ compared to 26.71% actual area coverage), and this is presented in the main text.”

I don’t understand why the authors are using 100 km² planning units for such a (relatively) small case study, where the dense population has led to small fragmented habitats that could be available for protection. The authors recognize that it’s unrealistic to expect 100% of land within these cells would be protected, but their solution is to expand the area target to 50% of cells. This is problematic for multiple reasons, as this avoids any actual reporting of the land available to be protected, and they don’t even report on the results of the 50% (except for its inclusion in the figure).

As a result, it is ambiguous just how much of the 30% of cells chosen for protection actually contain land that would be protected. Furthermore, they do not appear to exclude developed lands from potential selection. A cell chosen for protection could be 80% built-up areas and yet 100% of it contributes to their prioritized areas for protection.

The authors should remove developed land from consideration (unless with proper justification), and they should reduce the size of their planning units to more adequately reflect the highly fragmented state of UK habitats so they can more accurately estimate area coverage and assess true connectivity of their networks.

10x10km planning units are suitable for initial national analysis to locate areas to concentrate conservation funding. As we state in the manuscript, these units are the landscapes where: “The likelihood that individual threatened species will recover would be increased in these areas because they are already present, and thus available to colonise improved habitats that are delivered by upgraded protection and management in the surrounding landscape^{13,14}” (L147-150). Additionally we are using some data, i.e. BTO breeding bird atlas, which are only available at the 10x10km resolution.

The issue of land available has been addressed by the inclusion of costs in the additional analysis: here areas with more productive farmland and urban areas receive relatively higher cost weightings (L173-176, Table S5). The results are similar to the analysis without costs. Urban areas are still included, but with a high cost to inclusion. We believe this approach is appropriate as it is important to consider species complementarity of all areas (the government 30% target applies to the entire land surface) and some urban/near-urban areas do contain nationally rare species.

An important additional point we now include in the text is that different conservation action may have to be taken in different landscapes (L69-74): *“The priority cells recognised here represent foci for identifying and directing subsequent conservation actions and funding, accepting that different blends of conservation actions will be required in different landscapes. Given this constraint, an additional 50% coverage target is presented, within which a subset of higher-priority sites can be identified as foci for biodiversity and habitat ‘recovery’ in the wider countryside (Fig.1, Table S2). ”*

Additionally, the 50% coverage target results are now included in Table S2.

It is also strange that the authors do not consider range shifts amidst climate change. It is becoming very apparent that new protected areas need to protect not just what biodiversity features are present, but what biodiversity will be present in the future. This is becoming common practice in systematic conservation planning, yet the authors claim that protecting what’s “present” (Line 47) enhances the chances that species will recover and colonize the landscape. But that is not the optimal approach to securing species from extinction. The authors should consider adding expected shifts due to climate change, or provide good justification for why they are not considering it in their proposed protected areas.

We feel this proposed range shift analysis is beyond the scope of this short communication. We primarily wish to demonstrate that including designations in conservation targets arbitrarily is a highly inefficient strategy to conserve species. We have incorporated costs at the reviewers suggestion, we feel incorporating an additional climate change analysis (potentially under different scenarios) would dilute the central points of this communication.

3. There is a lack of reproducibility as the manuscript current stands. Since I have not received (nor seen any reference to) any supplementary material, I assume none exists. Without more detailed methods, I could not reproduce this analysis.

The authors do not state what species were included in the analysis. Did this include mammals, birds, insects, plants? There are also no references in the reference list to the species data. They mention the number of species with limited records, but they should provide a list of all species included in the analysis, and which ones were treated differently in the analysis.

We have now provided a complete list of all species included within the analysis. Details on taxa and exactly how they were included in the prioritisation are provided (Table S4).

The authors are too vague in their description of the analysis. For example they mention using a Frescalo analysis, where they apply weights for spatial proximity and habitat similarity (Line 138-140). How are the variables treated in the analysis, and what weights were applied? Simply stating that weights were applied does not tell me anything about how to reproduce it (or even verify if weights were applied properly). Another example in Line 136, the authors say means of weather data were used to calculate “relevant covariates”, but I have no idea what “relevant covariates” mean.

A supplementary workflow and methods section have been added, this gives more detail on the modelling process. The workflow has been added based on the comments of another reviewer to aid understanding of the approach as a whole.

Specifically with regards to Frescalo, slightly more informational text has been added in the Supplementary Methods and we more clearly direct the reader to a reference fully detailing how to undertake the analysis to estimate recorder effort.

“Frescalo works through a number of stages to estimate recorder effort, but see Hill (2012) for further details on use of Frescalo software to estimate recorder effort. Simply, for each cell a matrix of weights is created for neighbouring cells, with higher weights for spatial proximity and habitat similarity. Species presences are then multiplied by these weights, and recorder effort is estimated based upon the difference between the focal cell value and the neighbourhood mean cell value.”

With regards to covariates, the text has been changed to clarify we refer to previously mentioned covariates “A joint model of distribution intensity and recording effort was used, including biologically relevant covariates: seasonality, growing degree days, water availability, winter cold²³, and soil pH from the Countryside Survey 2007 dataset²⁴. These covariates were calculated from monthly means of weather data (mean temperature, sunshine and rainfall) for the decade to 2014 provided by the Met Office²⁵.”

The authors mention in Line 82 a random sample was taken, but in Line 175 they say 1000 random samples were drawn. Is it 1 or 1000 random samples? If it is the latter, the authors should also present the distribution of these results to compare with their scenarios. This is never mentioned in the Methods.

Previously 1000 samples were taken, but this has now been updated to 10 000. The text has been changed to clarify the calculation of the samples and a minimum and maximum values added.

L94-97: “This is only slightly more effective than undertaking scenario 2 by replacing landscape protection cells with the same number of randomly sampled cells (mean 33.01%, min 30.02%, max 35.93%; based on 10 000 iterations).

Reviewer #3 (Remarks to the Author):

Dear Editor,

The manuscript COMMSBIO-20-3548-T “Translating area-based conservation pledges into efficient biodiversity protection outcomes” is a short communication. It focuses on the issues that area-based conservation pledges without planning can bring to biodiversity conservation. It uses the case of United Kingdom to illustrate such issues. In UK there is a political commitment of protect 30% of the land by 2030 to support the recovery of nature. The UK’s government assumed to reach that 30% by dedicating 4% of new lands to the existing protected land network (26%). The authors claim however that such area-based pledges should be driven by the distribution of biodiversity (list of priority species). They shown how systematic conservation prioritization could improve the amount of biodiversity protected (species) and how it would result in a different conservation network regarding to nowadays. They combined advanced modelling tools to test a species-driven area-based conservation pledge under different land governance scenarios.

2. Overall impression of the work

The work is very pertinent for scientific community and in general to countries and society. Many countries have done area-based conservation pledges, but their efficiency in provide biodiversity protection outcomes remain poorly documented. The authors make an effort to advance discussion in this topic.

It is pleasant to read the manuscript. There is however space for improvements, namely by:

Main text

- The content is well organized.

- To access the numbers presented, the maps included may not be sufficient. The manuscript would be improved by including graph/summary statistics representations for the numbers presented (e.g. Line 84-92). These percentages are in base to what extent (e.g. km²).

This is now provided in Table S2, more completely:

Table S1 states the number or cells within each protection cut-off, i.e. how much of a cell must be protected for the cell to be considered ‘protected’. Only the 40% cutoff is presented in the main text

Table S2 lists the median species distribution protection for different scenarios, comparable to Fig 2, Fig 1a/b, and Fig S2 a/b.

Table S3 compares overlap between scenario 1 priorities and current landscape protection, comparable to Fig. 1c and Fig S2c

- What type of species are included in the work? Animals, plants? There is no indication about that. This information can help us to understand if the work covers the broad biodiversity or is for instance driven only by e.g. animal species. This information is relevant for the readers and for the overall statement “Translating area-based conservation pledges into efficient biodiversity protection outcomes”

Reviewer 1 also raised this issue - we have now provided a complete list of all species included within the analysis. Details on taxa and exactly how they were included in the prioritisation are provided (Table S4).

Methods

- Including a workflow showing the steps performed, assumptions made, input data used and connections between outputs. For a more common reader, this would benefit the understanding.

A supplementary workflow figure has been added with caption text that walks the reader through the approach. We believe this makes following the methods much easier for the reader.

- Adding additional details in the modelling performed could be also submitted as attachment to support the robustness and clearance of the modelling process.

We have added a Supplementary Methods section to address this comment with several equations detailing the spatial Bayesian models used.

- mentioning all datasets used (e.g. Where comes from the data used to select all cells with fractional land cover > 60%, considered later as suitable cells for conservation expansion.)

We have now added L143-45: *“All designation data used is publicly available from the respective national spatial data repositories for England¹⁸ (SSSI/NNR/NP/AONB), Scotland¹⁹ (SSSI/NNR/NP/NSA), and Wales²⁰ (SSSI/NNR/NP/AONB).”* with citations containing URLs to access the repositories.

- describing better in the methods the scenarios briefly introduced in the main text. They are important assumptions for the outcomes.

The scenarios are now described in the methods L167-72: *“However, priorities were constrained by masking or ‘locking in’ different relevant areas to each scenario such that all other cells must be removed first; reducing overall solution optimality but ensuring complementarity to masked areas. Scenario 1 only masked cells protected for biodiversity and didn’t consider other designations beyond that. Scenario 2 also masked cells protected for biodiversity but, corresponding to the 30by30 pledge, additionally masked protected landscapes.”*

3. *Specific comments, with recommendations for addressing each comment*

- 1- *Find and replace double spaces along the text.*

Removed where found.

- 2- *Please make some sentences shorter and more clear for general readers (e.g. line 76-80 "In terms of representativeness, both scenarios would protect more of the ranges of threatened species than the cells currently protected for biodiversity (median 1.67% distribution protected): the less constrained first scenario would ensure an additional 68.27% could be protected within 30% of cells in scenario 1, compared to 38.33% under scenario 2 (Fig 2)")*

We have attempted to simplify sentences throughout the manuscript where possible.

REVIEWERS' COMMENTS:

Reviewer #1 (Remarks to the Author):

I appreciate the authors' additional analyses, which have improved the robustness of their main analysis, as well as provided more insight into the planning of GB's 30by30.

There remain a few important points that have not been incorporated into the manuscript that would help clarify several aspects for the reader.

1. While I am still not entirely convinced that a 10x10km planning unit is optimal for such a highly fragmented country, the authors' additional statement "we are using some data, i.e. BTO breeding bird atlas, which are only available at the 10x10km resolution" is more compelling. I think the authors need to clarify in the main text that this 10km resolution is due to data availability (which is a very justified rationale), and if they are so inclined, they can then also argue that they believe this scale is also sufficient for purpose.

2. The authors in their rebuttal provide some useful commentary that doesn't appear to have made its way into the main text. Specifically: "Urban areas are still included, but with a high cost to inclusion. We believe this approach is appropriate as it is important to consider species complementarity of all areas (the government 30% target applies to the entire land surface) and some urban/near-urban areas do contain nationally rare species." This point should be cleaned and added to the manuscript to clarify why urban areas (which are often excluded from SCP analyses) are included.

3. On that note, seeing the list of species makes it clearer to me why including urban areas would still be meaningful. However, this information is relegated to the supplementary, and a reader of the main text still won't have an idea of what species were included. The authors should clarify somewhere in the main text (methods or maybe even earlier) that they are considering birds, terrestrial invertebrates, vascular and non-vascular plants. This will serve to (1) clarify the taxa relevant to this protected area network, and (2) add further justification for the incorporation of urban areas, which can be of high use to many of these species.

4. I understand that the incorporation of climate change in the planning design may be beyond the scope of this paper. That said, it is an exceptionally important point that must be considered in conservation planning and environmental policy. Given the policy relevance of this paper, I suggest the authors at least mention that climate change responses will be critical to monitoring and protecting threatened species into the future, beyond where they may be present today.

Reviewer #3 (Remarks to the Author):

Dear editor,

the manuscript "Translating area-based conservation pledges into efficient biodiversity protection outcomes" has been improved. Authors addressed adequately the concerns previously highlighted.

Minor comment: check and standardize the formatting of some of the references in the main text.

Translating area-based conservation pledges into efficient biodiversity protection outcomes

(Commun. Biol)

We thank both reviewers for their constructive and insightful comments on this manuscript, and we look forward to publishing this improved paper.

Comments are repeated in italics, and addressed in bold text either by a description of how they are adopted, or if not then an explanation. To illustrate how comments have been addressed, direct quotes are sometimes used from the manuscript.

REVIEWERS' COMMENTS:

Reviewer #1 (Remarks to the Author):

I appreciate the authors' additional analyses, which have improved the robustness of their main analysis, as well as provided more insight into the planning of GB's 30by30.

There remain a few important points that have not been incorporated into the manuscript that would help clarify several aspects for the reader.

1. While I am still not entirely convinced that a 10x10km planning unit is optimal for such a highly fragmented country, the authors' additional statement "we are using some data, i.e. BTO breeding bird atlas, which are only available at the 10x10km resolution" is more compelling. I think the authors need to clarify in the main text that this 10km resolution is due to data availability (which is a very justified rationale), and if they are so inclined, they can then also argue that they believe this scale is also sufficient for purpose.

We have now added the following text to the main text to highlight the spatial resolution limitations of the bird data used; L97 – “due to resolution of spatial data for certain taxa”, and L185-186 – “BTO bird atlas data are only available at the 10x10 km scale, which limited the spatial resolution of the analysis.”

2. *The authors in their rebuttal provide some useful commentary that doesn't appear to have made its way into the main text. Specifically: "Urban areas are still included, but with a high cost to inclusion. We believe this approach is appropriate as it is important to consider species complementarity of all areas (the government 30% target applies to the entire land surface) and some urban/near-urban areas do contain nationally rare species." This point should be cleaned and added to the manuscript to clarify why urban areas (which are often excluded from SCP analyses) are included.*

We agree this is important to mention, and have now included this point at the end of the methods when describing the opportunity cost layer. L214-217 - "Although urban areas are often excluded from SCP analyses, it is important to consider species complementarity of all landscapes (the government 30% target applies to the entire land surface). Since some urban/near-urban areas contain nationally rare species, we include urban areas, albeit imposing the maximum opportunity cost in these cells."

3. *On that note, seeing the list of species makes it clearer to me why including urban areas would still be meaningful. However, this information is relegated to the supplementary, and a reader of the main text still won't have an idea of what species were included. The authors should clarify somewhere in the main text (methods or maybe even earlier) that they are considering birds, terrestrial invertebrates, vascular and non-vascular plants. This will serve to (1) clarify the taxa relevant to this protected area network, and (2) add further justification for the incorporation of urban areas, which can be of high use to many of these species.*

Text highlighting the wide range of taxa included has now been added early on in the main text, L92-94 – "We identified the highest priority areas for network expansion that maximises coverage of 428 priority species distributions including birds, plants and a wide variety of invertebrates (Supplementary Data 1).

4. *I understand that the incorporation of climate change in the planning design may be beyond the scope of this paper. That said, it is an exceptionally important point that must be considered in conservation planning and environmental policy. Given the policy relevance of this paper, I suggest the authors at least mention that climate change responses will be critical to monitoring and protecting threatened species into the future, beyond where they may be present today.*

This sentence has been added in the summary paragraph, we agree it is important to incorporate this into any plan. LXXX – "Further development of priority conservation networks should consider how climate change will likely affect the distribution of species, habitats, and land use pressure^{19,20}, but securing the existing distributions of currently-threatened species remains a priority."

Reviewer #3 (Remarks to the Author):

Dear editor,

the manuscript "Translating area-based conservation pledges into efficient biodiversity protection outcomes" has been improved. Authors addressed adequately the concerns previously highlighted.

Minor comment: check and standardize the formatting of some of the references in the main text.

References standardised as much as possible.